# A Structural Equation Model of Achievement Emotions, Coping Strategies and Engagement-Burnout in Undergraduate Students: A Possible Underlying Mechanism in Facets of Perfectionism

**DOI:** 10.3390/ijerph17062106

**Published:** 2020-03-22

**Authors:** Jesús de la Fuente, Francisca Lahortiga-Ramos, Carmen Laspra-Solís, Cristina Maestro-Martín, Irene Alustiza, Enrique Aubá, Raquel Martín-Lanas

**Affiliations:** 1School of Education and Psychology, University of Navarra, 31009 Pamplona, Spain; 2School of Psychology, University of Almería, 04120 Almería, Spain; 3Department of Psychiatry and Clinical Psychology, University Clinic of Navarra, 31008 Pamplona, Spain; flahortiga@unav.es (F.L.-R.); claspra@unav.es (C.L.-S.); cmaestro@unav.es (C.M.-M.); ilalustiza@unav.es (I.A.); eauba@unav.es (E.A.); rmlanas@unav.es (R.M.-L.)

**Keywords:** achievement emotions, coping Strategies, engagement-burnout, university students, perfectionism

## Abstract

Achievement emotions that the university student experiences in the learning process can be significant in facilitating or interfering with learning. The present research looked for linear and predictive relations between university students’ achievement emotions, coping strategies, and engagement-burnout, in three different learning situations (classroom, study time, and testing). Hypotheses were identified for a possible model that would analyze the two facets of perfectionism based on these relations. In the case of perfectionistic strivings, the test hypothesis was that positive emotions would predispose the use of problem-focused coping strategies and an emotional state of engagement; in the case of perfectionistic concerns, however, negative emotions would predispose the use of emotion-focused strategies and a state of burnout. A total of 654 university students participated in the study, using an online tool to complete validated questionnaires on the three study variables. All students provided informed consent and corresponding permissions. Given the ex-post facto linear design, the predictions could be verified for each situation by means of logistic regression analyses and Structural Equations Models (SEM). Empirical results lent support, in varying degree, to the proposed theoretical relations. The testing situation was of particular interest. We discuss implications for perfectionism research and for the practice of prevention, education and health care in the university setting.

## 1. Introduction

The psychological well-being of college students is increasingly recognized as an important concern within higher education. The study of students’ emotional experiences in the teaching-learning context has produced a great deal of research on aspects not previously considered under the cognitivist paradigm. Such research seeks to explain to what degree emotional processes facilitate or interfere in learning processes [1,2,3,4,5,6,7]. Specifically, the level of stress experienced by students who try to meet the demands and requirements of university study has captured the interest of researchers [8,9]. 

When considering academic stress, Clinical and Health Psychology give research priority to individual predictive or explanatory factors, such as personality variables, anxiety, and cognitive differences [10,11]. From the Educational Psychology perspective, however, academic stress can be considered a contextualized phenomenon within the learning process [12], especially in formal, high-pressure contexts. Classroom, study and testing situations represent an increasing progression of stressful stimuli, requiring college students to manage their emotions on a daily basis [13]. Emotions that are produced in these situations have a determining influence on students’ well-being and achievement.

Depending on their individual characteristics, students in these contexts use different methods or coping strategies to manage stress [14]. Prior research has reported predictive relationships between self-regulation and coping strategies [15], and between resilience, coping and burnout [16]. However, the predictive relationship between achievement emotions and coping strategies has yet to be clearly established, as well as the effect of these two variables on the motivational state of engagement versus burnout in university students. This is therefore the focus of the present investigation. This relationship, moreover, may also be important in helping to clarify emotional mechanisms that are involved in the different types of perfectionism.

## 2. Perfectionism as a Personal Academic Variable

Academic perfectionism can be described as setting exceedingly high standards, then pursuing those standards with relentless self-criticism [17]. Reactivity to stress also seems to be influenced by perfectionism. Perfectionism as a multidimensional construct -- with adaptive and maladaptive facets—has been supported by multiple studies over the past twenty years [18]:

1) Personal Standards Perfectionism (PSP), or perfectionistic strivings, is considered an adaptive aspect of perfectionism. The individual sets and pursues high standards and goals, a practice that has been associated with psychological wellbeing, as indicated by adaptive aspects including task-enjoyment, positive affect, and satisfaction [19]. 

2) Evaluative Concerns Perfectionism (ECP), or perfectionistic concerns, by contrast, constitute a maladaptive aspect of perfectionism. Unrealistically high standards and expectations are followed by overly critical self-assessment, negative reactions to failure, and preoccupation with criticism and expectations from others. Several studies find perfectionistic concerns to be associated with maladaptive indicators of psychological well-being: depression, distress, anxiety, and hopelessness [20,21]; reduced well-being, depression, burnout and anxiety [22,23], concern over mistakes, doubts about actions, socially prescribed perfectionism, discrepancy, and negative reactions to imperfections [24].

### 2.1. Achievement Related Emotions as an Affective Variable of the Learning Process

Achievement emotions, an affective component of learning, have become a specific topic of research interest. Based on Pekrun’s control-value theory [25,26], achievement emotions are determined by the interaction of two components: the perceived controllability of achievement activities and their outcomes, and value appraisals of the subjective value or importance of these activities or outcomes. Academic emotions, more broadly, include achievement emotions experienced in an academic context, as well as any emotions related to (1) the instruction, (2) the study process, or (3) an exam situation [27,28,29,30,31]. These three types of situations are considered representative of the three levels of academic stress experienced by college students [25].

Pekrun [32] went beyond previous conceptualizations [33,34] to classify academic emotions along three axes: their focus, valence, and activation (for an overview, see [35]). The source (focus) of academic or achievement emotions can be either the: (a) activity, relating to ongoing activities involved in achieving, or the (b) outcome, pertaining to concerns about achievement outcomes [25]. For both activity and outcome emotions, their valence can be either positive or negative (pleasant or unpleasant), and their role in *activation* either activating or deactivating. Recent research addresses certain activity emotions in academic settings, for example: the positive, activating emotion of enjoyment (for an overview, see [26]) and the negative, deactivating emotion of boredom (for an overview, see [27]). In general, we assume that positive activating emotions (enjoyment, hope, pride) have a positive impact on achievement, while negative emotions (anger, anxiety, shame, hopelessness), and deactivating emotions (boredom, relief) negatively affect achievement and learning behavior. Empirical evidence supports this assumption, in classroom, study and exam situations [28,29].

Recent research has supplied plentiful evidence on the role of achievement emotions in the university context [30]. Positive activating emotions (enjoyment, hope, pride) were reported to be interrelated with metacognitive monitoring processes in multimedia learning tasks, but negative emotions (frustration) and deactivating emotions (boredom) have been shown to negatively predict self-monitoring [31]. Elsewhere, the negative impact of test anxiety has been verified, and potential control mechanisms have been explored [32]. The effect of rumination on university students’ negative affect and on their achievement has also been confirmed [33].

While research findings increasingly identify the specificities of academic emotions, there has been little attempt to search out the underlying mechanisms. Relationships between negative emotions and emotion-focused coping strategies have been reported in secondary education [34], but not at university. The present study contributes evidence in this direction, exploring how academic emotions in university students relate to their stress coping strategies; how emotions and coping strategies together affect motivational state (engagement versus burnout) and how they may also be an underlying mechanism in the two facets of perfectionism.

### 2.2. Coping Strategies as a Meta-emotional Variable of the Learning Process

Coping strategies are a psychological construction referring to strategic knowledge, skills and behaviors that people use to manage emotions in a given situation; they are thus conceptualized as meta-emotional skills [35]. Categorizations of coping vary substantially among researchers and theoretical orientations. In general, coping strategies tend to be grouped into categories according to the degree that the strategies are beneficial/adaptive or detrimental/maladaptive. Lazarus and Folkman [36] proposed an initial categorization of coping strategies that identifies two types of focus: (1) emotion-focused strategies that seek to manage, minimize or avoid negative emotional states (distraction, reducing anxiety, preparing for the worst, emotional venting, resigned acceptance); and (2) problem-focused strategies that manage or reduce the causes of the stressful experience or of overextended personal resources (help-seeking, self-instructions, positive reappraisal, social support, alternative reinforcement). While the first version of the Cognitive Theory of Stress and Coping [37] assumes that the state of stress is associated with negative emotions, the Revised Stress and Coping model [38] adopts the position that positive emotions and negative emotions co-occur in stress states [39].

Several studies have tried to identify strategies as being adaptive or maladaptive. Adaptive and maladaptive strategies have been identified in the literature on critical incident, traumatic stress, and occupational stress [40], including such examples as anger, distancing, planned effort, positive reappraisal, and social support [41], and maladaptive avoidant and ruminative coping [42]. 

Prior research on motivational-affective factors in university learning has also stressed the importance of particular aspects of how university students cope with stress: religious coping (Francis et al., 2018); the role of health habits as a coping strategy [43]; coping in relationship to well-being [44,45,46]. Also, have analyzed the predictive role of achievement emotions in coping strategies and in motivational states of engagement-burnout [47].

An analysis of predictive relations between academic emotions and coping strategies can help us establish mechanisms that then relate these to the motivational states of engagement or burnout. This relationship, in addition to its own relevance, could later be incorporated into models of positive and negative aspects of perfectionism, as described in sections above.

### 2.3. Engagement-Burnout as a Motivational Variable of the Learning Process

The constructs of engagement and burnout are motivational-affective in nature and refer to a student’s emotional state in an instructional context. The two constructs can be considered polar opposites, representing two extremes of the same aspect. While burnout represents fatigue, depersonalization, lack of expectations and disaffection for one’s work [48], engagement represents a liking for, engagement and enjoyment of one’s work [49]. Some recent studies express doubt as to whether the “engagement-burnout” construct constitutes a single dimension that goes from commitment (implication) to wear (attrition). Leiter and Maslach [50,51] themselves point out that the two constructs may be related, while not absolutely opposite of each other.

Previous research has reported factors that predict and predispose both constructs [52]. Achievement emotions (positive vs. negative) have been differentially associated with burnout [53], and engagement has been shown to favor metacognitive self-regulation and knowledge construction [54]. More recently, the duality has been conceptualized as positive learning or engagement vs. negative learning or burnout [55]. The importance of engagement has also been reported in service-learning situations at university [56]. Burnout, for its part, has consistently appeared as a negative predictor of motivation and achievement [57,58]; however, the inventory authors have recognized that the two constructs have a complex relationship, requiring more specific analyses by profiles [50,51]. Understanding the relationship of burnout-engagement to university students’ achievement emotions and coping strategies would make it possible to assess the profile of these factors in the two facets of perfectionism.

## 3. Aims and Hypotheses

The existing theoretical models of learning-related emotions have not considered that types of achievement emotions (emotional variables) may be associated with types of coping strategies (meta-emotional variables), and that this may affect university students’ state of engagement-burnout (motivational variable). A hypothetical relational model that incorporates achievement emotions, coping strategies, motivation, and facet of perfectionism is represented in Figure 1. Specifically, (1) ECP, representing negative reactions to failure and being overly self-critical, is typically associated with maladaptive outcomes, such as reduced well-being, depression, burnout and anxiety [22]; while (2) PSP, representing the setting of high standards and goals, is associated with adaptive indicators of psychological wellbeing, such as task-enjoyment, positive affect, and satisfaction [19].

Based on this proposed model, the aim of this research was to verify any linear, predictive relations between achievement emotions (emotional variable), coping strategies (meta-emotional variable) and engagement-burnout (motivational variable) in university students, during three different learning situations: the classroom, study time, and testing. Predictive, structural, linear relationships would allow us to verify the direct and indirect effects of certain variables on others. Specifically, they provide an explanatory, mediational empirical model of coping strategies with respect to the other two variables in each specific situation (class, study and testing). This represents a methodological advance, given that the predictive relationship, direct and indirect, cannot be identified using classic variance analyses. Consequently, we established these hypotheses: (H1) positive emotions will predispose the use of problem-focused coping strategies and an attitude/emotional state of engagement when learning; (H2) negative emotions will predispose the use of emotion-focused strategies, and an emotional state of burnout when learning.

## 4. Method

### 4.1. Participants

Drawing from the two universities participating in this research project, a convenience sample was formed of students who completed the questionnaires. There were 642 undergraduate students from the two Spanish universities, and ten teaching and learning processes (from ten academic subjects) were assessed. The sample was composed of students enrolled in Psychology and Primary Education degree programs; 83.5% were women and 16.5% were men. Their ages ranged from 19 to 45 years, with a mean age of 20.13 (sd = 5.8) years. Participation was anonymous and voluntary. The Guidance Department at each university extended an invitation to participate to the teachers in the relevant departments, and the participating teachers offered the invitation to their students. Teacher and student participation was recognized with the Certificate of Participation in an R&D Project. Each academic subject (specific teaching-learning process) was assessed through online questionnaires.

### 4.2. Instruments

*Achievement Emotions.* The Achievement Emotions Questionnaire, AEQ [29] is a multi-dimensional self-report instrument that assesses achievement emotions in university students. The questionnaire was generated as part of a quantitative and qualitative research program that analyzed emotions experienced by students in academic achievement situations (for a summary, see [13]). Several discrete emotions are measured in the context of the three main situations pertaining to academic achievement: attending class, studying, and completing tests and exams. The AEQ in its current version measures eight class-related emotions, eight study-related emotions, and eight emotions during testing. Thus, the three sections of the AEQ correspond to these situations of classroom, study time, and testing. The class-related emotions scale (CRE) contains 80 items that measure the following eight emotions: class-related enjoyment, hope, pride, anger, anxiety, shame, hopelessness, and boredom. The learning-related emotions scale (LRE) uses 75 items to measure the same eight emotions in study situations. The test emotions scale (TES) contains 77 items for assessing test-related enjoyment, hope, pride, relief, anger, anxiety, shame, and hopelessness. Each section contains three sub-sections that address the emotions felt before, during and after the academic situation covered by that section. The student’s trait achievement emotions are assessed, in other words, his or her typical personal emotional reactions to achievement situations. Instructions for the AEQ can be modified in order to measure emotions experienced in a particular class subject (course-specific emotions), or in specific situations at specific moments (state achievement emotions).

The AEQ measures four positive emotions (enjoyment, hope, pride, and relief) and five negative emotions (anger, anxiety, hopelessness, shame, and boredom). Two main criteria were used for deciding what emotions to include. First, emotions frequently experienced by college students were identified [25]. Second, the emotions were classified along two dimensions, each having two possible values: valence (positive vs. negative) and activation (activating vs. deactivating) (see [53,54]). Four categories of emotions result from the combination of these values, and reflect how emotions affect learning, achievement, personality development, and health. The resulting categorizations are: positive activating: enjoyment, hope, pride; positive deactivating: relief; negative activating: anger, anxiety, shame, hopelessness; negative deactivating: boredom.

There are differences in the function and social structure of the three basic types of university achievement situations (attending class, studying, and taking tests). Consequently, the emotions experienced in these situations also differ. For example, enjoying classroom instruction is not the same as enjoying the challenge of an exam. While some students may feel excited about going to class, others feel excited when facing a test. This is taken into account in the AEQ through separate scales for emotions relating to the class setting, study time, and testing:

1) Class-Related Emotions (translation: [55]). CRE psychometric properties were found to be satisfactory in students from Spain. The model obtained good fit indices in this sample. Also verified were unidimensionality of the scale and metric invariance in the samples evaluated (Chi Square = 10885.597, Degrees of freedom = 3052, *p* < 0.001; CFI = 0.951, TLI = 0.952, IFI = 0.963, TLI = 0.958, and CFI = 0.952; RMSEA = 0.041; HOELTER = 458, *p* < 0.05; 466 *p* < 0.01). Cronbach’s alpha for this sample was 0.904, with 0.803 (40 items) and 0.852 (40 items) for the two parts, respectively (80 items).

2) Learning-Related Emotions (translation: [56]). LRE psychometric properties were found to be satisfactory in students from Spain. The model obtained good fit indices in this sample. Also verified were unidimensionality of the scale and metric invariance in the samples evaluated (Chi Square = 10885.597, Degrees of freedom = 3052, *p* < 0.001; CFI = 0.959, TLI= 0.942, IFI= 0.969, TLI= 0.955, and CFI = 0.958; RMSEA = 0.038; HOELTER = 501, *p* < 0.05; 511 *p* < 0.01). Cronbach’s alpha for this sample was 0.930, with 0.880 (38 items) and 0.846 (37 items) for the two parts, respectively (75 items).

3) Test-Related Emotions (translation: [56]). TRE psychometric properties were found to be satisfactory in students of Spain. The model obtained good fit indices in this sample. Also verified were unidimensionality of the scale and metric invariance in the samples evaluated (Chi Square = 10885.597, Degrees of freedom = 3052, *p* < 0.001; CFI = 0.954, TLI = 0.946, IFI = 0.964, TLI = 0.959, and CFI = 0.953; RMSEA= 0.039; HOELTER = 492, *p* < 0.05; 502 *p* < 0.01). Cronbach’s alpha for this sample was 0.913, with 0.824 and 0.869 for the two parts, respectively (77 items). 

*Coping Strategies (meta-emotional variable)*. To measure coping strategies, we used the EEC-Short [57], a short, validated Spanish version of the *Coping Strategies Scale*, EEC [58]. While the original instrument contained 90 items, the validation produced a first-order structure of 64 items and a second order with 10 factors and two significant dimensions, the latter having adequate fit values [(Chi-square = 878.750; Degrees of freedom (77-34) = 43, p < 0.001; NFI = 0.901; RFI = 0.945; IFI = 0.903, TLI = 0.951, CFI = 0.903, RMSEA= 0.07]. For reliability measures, Cronbach alpha values were 0.93 (complete scale), 0.93 (first half) and 0.90 (second half), Spearman-Brown was 0.84 and Guttman was 0.80. Two dimensions are evaluated: D1. Emotion-focused coping (0.95); D2. Problem-focused coping (0.91). The emotion-focused strategies were: F1. Avoidant distraction (0.79); F7. Reducing anxiety and avoidance (0.88); F8. Preparing for the worst (0.80); F9. Emotional venting and isolation (0.91); and F10. Resigned acceptance (0.86). Problem-focused strategies were: F2. Seeking family help and counsel (0.92); F5. Self-instructions (0.82); F10. Positive reappraisal and firmness (0.87); F12. Communicating feelings and social support (0.89); and F13. Seeking alternative reinforcements 0.80). See Table 1.

*Engagement-Burnout.* Cross-cultural studies have shown adequate reliability and construct validity indexes for this construct. A validated Spanish version of the Utrecht Work Engagement Scale for Students [59] was used to assess Engagement. The psychometric properties were satisfactory in students from Spain. The model obtained good fit indices in this sample, with a second-order structure of three factors: vigor, dedication and absorption. Also verified were unidimensionality of the scale and metric invariance in the samples evaluated [Chi Square = 792.526, *df* = 74, *p* < 0.001; CFI = 0.954, TLI= 0.976, IFI= 0.954, TLI= 0.979, and CFI= 0.923; RMSEA= 0.083; HOELTER = 153, *p* < 0.05; 170 *p* < 0.01]. Cronbach’s alpha for this sample was 0.900 (14 items), with 0.856 (7 items) and 0.786 (7 items) for the two parts, respectively. 

The validated Spanish version of The Marlach Burnout Inventory, MBI [60] was also used to assess Burnout. Psychometric properties for this version were satisfactory in students from Spain. The model obtained good fit indices in this sample, with a second-order structure of three factors: exhaustion or depletion, cynicism, and lack of effectiveness. Also verified were unidimensionality of the scale and metric invariance in the samples evaluated [Chi Square = 767.885, *df* = 87, *p* < 0.001; CFI = 0.956, TLI= 0.964, IFI= 0.951, TLI= 0.951, and CFI = 0.953; RMSEA = 0.071; HOELTER = 224, *p* < 0.05; 246 *p* < 0.01]. Cronbach’s alpha for this sample was 0.874 (15 items), with 0.853 (8 items) and 0.793 (7 items) for the two parts, respectively. 

### 4.3. Procedure

Participants voluntarily completed the scales using an online platform [61] [http://www.estres.investigacion-psicopedagogica.com/english/seccion.php?idseccion=1]. All students gave their informed consent through an online signature that is required when creating an account on the platform, before any questionnaires are completed. Ten specific teaching-learning processes were evaluated, covering different university subjects over a two-year period. To avoid fatigue, students were asked to complete just one questionnaire at a time, at two different times each week, over a four-month period. They were awarded a Certificate of Participation in Research as an incentive to maintain their motivation and recognize their effort. Presage variables (personality and others) were evaluated in September-October of 2017 and of 2018, Process variables (Academic Emotions) in February-March of 2018 and of 2017, and Product variables (Coping Strategies, Engagement-Burnout) in May-June of 2017 and of 2018. The procedure was approved by the respective Ethics Committees (ref. 2018.170), in the context of an R & D Project (2018-2021).

### 4.4. Data Analysis

In order to address the objectives and linear hypotheses, we used an ex post facto design for linear (noncausal) prediction. The hypotheses were tested using (1) multiple linear regression analysis and (2) three SEM (Structural Equation Models) analyses. In both cases, they were tested for each of the three situations of stress: classroom, study time and testing. The database had initially been reviewed and any incomplete cases were eliminated. (1) Multiple regression analysis was conducted using SPSS (IBM; v.25.0). Bivariate correlational analyses were not carried out, as they are more limited when establishing multiple linear prediction. (2) Structural validity analysis was conducted using AMOS (v. 23.0) for Windows, as was construction of the structural prediction model, specifically, verification of the structural linear prediction hypothesis (path analysis). The Comparative Fit Index (CFI) and the Root Mean Square Error of Approximation (RMSEA) were used to interpret the confirmatory factor analysis (CFA) and fit of the structural equation model (SEM). CFI values were used to identify acceptable and close fit to the data, namely, values equal to or more than 0.90 and 0.95, respectively [62]. RMSEA values equal to or below 0.05 and 0.08, respectively, were taken to indicate close and acceptable levels of fit [63]. [Research has identified cutoff points in the form of beta coefficients for qualifying direct effects: less than 0.05 is considered too small to be meaningful, above 0.05 is small but meaningful, above 0.10 is moderate, and above 0.25 is large [64]. For indirect effects, we used Kenny’s definition [65] of an indirect effect as the product of two effects. Following Keith’s benchmarks, we proposed an educationally meaningful, small indirect effect = 0.003, moderate = 0.01, and large = 0.06.

## 5. Results

### 5.1. Linear Predictive Relationships 

The multiple regression analyses showed different significant relationships between achievement emotions, coping strategies and engagement-burnout attitudes in each situation:

#### 5.1.1. Classroom Situation

In the classroom situation, overall, positive emotions were a statistically significant predictor of problem-focused strategies, while negative emotions predicted emotion-focused strategies. In the case of positive emotions, the emotion of hope was particularly powerful in negatively predicting strategies F9 (Emotional venting and isolation) and F11 (Resigned acceptance) and positively predicting all the problem-focused strategies, especially strategy F10 (Positive reappraisal and firmness). The emotion of pride predicted certain problem-focused strategies like F12 (Communicating feelings and social support) and F13 (Seeking alternative reinforcement). In the case of negative emotions, the emotion of anger positively predicted strategy F9 (Emotional venting and isolation) and negatively predicted problem-focused strategies, such as F2 (Seeking help and family advice) and F12 (Communicating feelings and social support). In addition, the emotion of boredom, as a negative, deactivating emotion, had power for predicting emotion-focused strategies in the case of F1 (Avoidant distraction), F7 (Reducing anxiety and avoidance) and F8 (Preparing for the worst). However, the emotion with the greatest predictive power was anxiety, with predictions similar to those of boredom, favoring the use of emotion-focused strategies F1, F7 and F8, but also problem-focused strategies F2 and F12. Shame significantly predicted strategies F9 (Emotional venting and isolation) and F11 (Resigned acceptance). For its part, hopelessness negatively predicted F7 (Reducing anxiety) and F5 (Self-instructions). See Table 2.

#### 5.1.2. Study Situation

In the study situation, positive emotions positively predicted the use of all problem-focused strategies and certain emotion-focused strategies (F1 and F8), while negative emotions were positive predictors of all emotion-focused strategies and inversely predicted certain problem-focused strategies (F2 and F13). Specifically, the emotion of enjoyment positively predicted strategies F5 (Self-instructions) and F10 (Positive reappraisal and firmness). The emotion of hope significantly and positively predicted most problem-focused strategies and negatively predicted the emotion-focused strategy F11 (Resigned acceptance). The emotion of pride positively predicted strategies relating to social support, focused on the problem, as in F2 (Seeking help and family advice) and F12 (Communicating feelings and social support), as well as focused on emotion, as in F8 (Preparing for the worst). Also in this situation, the negative deactivating emotion of boredom was a strong negative predictor of problem-focused strategies F2 (Seeking help and family advice) and F10 (Positive reappraisal and firmness), and positive predictor of the emotion-focused strategy F7 (Reducing anxiety and avoidance). The negative emotion predicting the second highest number of strategies was anxiety, which predicts both problem-focused strategies (F7, F8) and emotion-focused strategies (F2, F12, F13). However, shame predicted only emotion-focused strategies (F8, F9, F11). See Table 2.

#### 5.1.3. Testing Situation

In the testing situation, while positive emotions predicted the use of problem-focused strategies, negative emotions predicted both emotion-focused strategies and problem-focused strategies, although predictive strength was greater in the emotion-focused strategies. One effect not observed in the other situations was that enjoyment significantly predicted the emotion-focused strategy F9 (Emotional venting and isolation). Also in this situation, the positive emotion hope positively and significantly predicted most problem-focused strategies (F10, F4, F12, F13), while pride showed less positive predictive power (F12, F2). The negative, deactivating emotion of boredom had no predictive value in this situation. The negative emotion of anger negatively predicted strategy F10 (Positive reappraisal and firmness), but positively predicted the emotion-focused strategy F9 (Emotional venting and isolation), as well as certain problem-focused strategies (F12 and F13). The negative emotion of anxiety proved to be a positive, significant predictor of strategy F11 (Resigned acceptance) and F2 (Seeking help and family advice). Finally, the emotions shame and hopelessness were predictors of F1 (Avoidant distraction) and F8 (Preparing for the worst).

One important effect to note is that in all three situations, total positive achievement emotions were a positive, significant predictor of problem-focused coping strategies F5 (Self-Instructions) and F10 (Positive reappraisal and firmness), while total negative achievement emotions were a positive, significant predictor of strategy F9 (Emotional venting and isolation). See Table 2.

## 6. Structural Prediction Relationships

### 6.1. Multivariate Relation Pathway: Class Situation (Stress Level 1)

The results of pathway analysis (SEM) showed an acceptable model of the relationship between variables. The relationship parameters of both models are set out below. Two models were tested; the second obtained more consistent results and was taken as definitive. See Table 3.

*Standardized Direct Effects.* This predictive linear model establishes that lack of positive emotions (POS) negatively predicted (−0.28) problem-focused strategies (PROB), and was a negative predictor of (0.20) engagement (ENG); this lack also positively predicted (0.52) negative emotions (NEGAT) and burnout (0.24) (BURN). Negative emotions (NEGAT) more strongly predicted (0.56) emotion-focused strategies (EMOT), which in turn were positive predictors (0.26) of burnout (BURNT); they were also negative predictors (-0.44) of engagement (ENG). Consequently, burnout (BURNT) was predicted by an absence of positive emotions (POS) and the presence of negative emotions (NEGAT), in conjunction with emotion-focused strategies (EMOT). See Table 4.

*Standardized Indirect Effects.* The model also contributed *multiple indirect* predictions among the variables. Complementing the direct effects, lack of positive emotions also had indirect predictive effects on emotion-focused coping (0.295), negative effects on engagement (−0.288) and positive on burnout (0.292). The absence of positive emotions positively predicted numerous negative emotions and emotion-focused coping strategies. Negative emotions also predicted coping strategies focused on emotion. Coping strategies focused on the problem positively predicted *engagement*, while emotion-focused strategies predicted burnout. See Table 5. 

*Graphic representation of the structural model*. The final model is graphically represented in Figure 2. 

### 6.2. Study Situation (Stress Level 2)

*Standardized Direct Effects.* In this situation, as in the previous, absence of positive emotions negatively (-0.35) predicted problem-focused strategies and was a negative predictor or engagement (0.20); this absence also positively predicted negative emotions (0.52) and burnout (0.24). For their part, negative emotions more strongly predicted (0.56) emotion-focused strategies (0.56), which in turn were positive predictors (0.26) of burnout; they were also negative predictors (-0.44) of engagement. Consequently, burnout was predicted by an absence of positive emotions and the presence of negative emotions, in conjunction with emotion-focused strategies.

The results of pathway analysis (SEM) showed an acceptable model of the relationship between variables. The relationship parameters of both models are set out below. Two models were tested; the second obtained more consistent results and was taken as definitive. See Table 6 and Table 7.

*Standardized Indirect Effects.* The model also contributed the existence of multiple indirect predictions among the variables. Complementing the direct effects, lack of positive emotions also had indirect predictive effects on emotion-focused coping (.306), negative effects on engagement (-0.314) and positive on burnout (.306). The absence of positive emotions positively predicted numerous negative emotions and emotion-focused coping strategies. Negative emotions also predicted coping strategies focused on emotion. Coping strategies focused on the problem positively predicted *engagement*, while emotion-focused strategies predicted burnout. See Table 8 and Figure 3. 

### 6.3. Test Situation (Stress Level 3)

The results of pathway analysis (SEM) showed an acceptable model of the relationships between variables. The relationship parameters of both models are set out below. Two models were tested; the second obtained more consistent results and was taken as definitive. See Table 9 and Figure 4.

*Standardized Direct Effects*. In this situation, unlike the two previous ones, the presence of positive emotions positively predicted (0.21) problem-focused strategies, which in turn were positive predictors of engagement (0.20); they were also negative predictors of negative emotions (-0.52) and burnout (-0.24). For their part, negative emotions more strongly predicted (0.56) emotion-focused strategies (0.56), which in turn were predictors (0.26) of burnout; they were also negative predictors (-0.44) of engagement. Consequently, engagement was predicted by positive emotions and problem-focused strategies, while burnout was predicted by negative emotions and emotion-focused strategies. See Table 10.

*Standardized Indirect Effects*. The model also revealed multiple indirect predictions among the variables. Complementing the direct effects, positive emotions also had indirect predictive effects on emotion-focused coping (-0.236), positive effects on engagement (0.227) and negative on burnout (-0.246). Positive emotions negatively predicted numerous negative emotions and emotion-focused coping strategies. Negative emotions also predicted coping strategies focused on emotion. Problem-focused coping strategies positively predicted engagement, while emotion-focused strategies predicted burnout. See Table 11. 

## 7. Discussion

The objective of this investigation was to contribute new evidence to verify whether positive versus negative emotions predicted problem- or emotion-focused coping strategies, and ultimately, a state of engagement versus burnout, in the context of perfectionism research. This intriguing objective took the shape of two hypotheses which were in large measure validated. 

Hypothesis 1 -positive emotions would predispose the use of problem-focused coping strategies and an attitude or emotional state of engagement when learning- was confirmed in the all situation by linear regression analyses and structural multiple prediction analyses (in the exam situation). Positive emotions consistently predicted problem-focused coping strategies (F5: Self-instructions; and F10: Positive reappraisal and firmness) and engagement. Regarding Hypothesis 2 -negative emotions were found to predict unhealthful coping strategies (F9: Emotional venting and isolation), and an ultimate state of burnout-, was confirmed in all three situations by linear regression analyses and structural multiple prediction analyses.

These results are consistent with prior evidence that has established associations between positive emotionality and self-regulation, and between negative emotionality and a lack of self- regulation [47,48,49,50,51,52,53,54,55,56,57,58,59,60,61,62,63,64,65,66]. Results presented in this research study also reveal the specific strategic coping behaviors that are associated with students’ emotions and their ultimate emotional state of engagement vs. burnout, suggesting that coping strategies are a meta-emotional variable, or, as their name implies, a behavioral strategy, adjusted or maladjusted, for managing emotions. Although the revised model by Lazarus and Folkman [36] already notes that the use of both types of strategies can be adaptive in daily life, the present study suggests a specific explanatory mechanism for the two types: (1) positive emotions, typical of the absence of stress, are most likely to result in the use of problem-focused strategies, with less focus on emotions, which in turn will give rise to a state of engagement; (2) negative emotions, typical of stressful states, would result in a preference for emotion-focused strategies, which ultimately would lead to burnout, and (3) the absence of positive emotions would also lead to burnout. These results seem to suggest that coping strategies, besides acting as a mediating variable, also act directly, along with emotions, in producing one state or another. For this reason, coping strategies could be considered a more precise behavioral mechanism in engagement vs. burnout. It seems plausible that the psychological wear and tear of negative achievement emotions (or the absence of positive ones), together with the strategic effort involved in managing them through extensive use of emotion-focused strategies, can result in burnout.

When considering each academic situation, however, the proposed predictive structural model (SEM) was not the exact model found in the classroom and study situations, due to the negative weight of positive emotions. That is, a lack of positive emotionality --associated with the presence of negative emotionality-- produced a preference for using emotion-focused strategies to cope with the emotional state of burnout during class and during study time. This result is highly interesting, because it suggests that the teaching process is involved. Although the teaching process is not the object of this analysis, recent evidence shows that teaching does play its role, and university students’ emotional state is the combined product of students’ personal characteristics and contextual characteristics [66,67]. The proposed model, however, was fulfilled in the testing situation. These differential results would indicate that achievement emotions operate differently in the three situations to which university students are exposed, as analyzed here. The causes of these differences could be the object of future research. The stress factors implicit to certain teaching processes would probably be key to understanding these results (de la Fuente, et al., 2020). In any event, the analysis of achievement emotions has added factors not traditionally considered onto the agenda of university research [68,69].

## 8. Limitations and Future Research

This evidence leads us to ask: why do the university students assessed have such little positive emotionality (enjoyment, confidence and pride) in classroom and study situations, and why is it that positive emotionality is activated in a testing situation? Is the problem with teaching or with learning? Does it depend on personality or gender variables? These factors have not been the object of study here, and therefore represent a study limitation. Our hypotheses have been partially supported by the results presented here. Future studies should follow the direction or trend established here with more precise methodology. The role of certain variables should be clarified, especially those referring to the role of the teaching process, in its interaction with the personal variables analyzed in the present study [70]. Despite these limitations, this study has been able to connect variables that had not been sufficiently analyzed to date, namely, achievement emotions (affective variables of the learning process), coping strategies (meta-emotional variables of the learning process) and emotional state (motivational variable of learning). 

The limited number of degree programs represented in our participants, and the total number of participating students, also represent important limitations. In order to make generalizable inferences, future studies must expand the number of degree programs represented and the international profile of participating students. Taking all this into account, the results found here are meaningful but must be taken with caution.

Another limitation pertains to the use of a linear predictive methodology. An associative type methodology does not allow us to infer causality or interdependence, only probability prediction. Future research should clarify the papel of age and gender in the relationships found. Additionally, future research should explicitly analyze this plausible, hypothesized relationship, not explicitly verified in the present research report. This line of analysis could help us better understand the differences between types of perfectionism. Previous evidence has generally reported a negative association between perfectionistic strivings and burnout symptoms such as exhaustion, cynicism, and inefficacy. On the other hand, some researchers [71] found the opposite effect, reporting that perfectionistic strivings correlated positively to exhaustion and cynicism. Perfectionistic concerns involve a persistent, negative reaction to imperfections, thereby increasing fatigue [72], and burnout [73]. One possible reason for this is that negative emotions and the effort required to manage them involve the use of many emotional and cognitive resources, which in turn leaves fewer resources for pursuing problem-focused strategies, hence resulting in ego-depletion and burnout. 

## 9. Implications and Applicability

Understanding how achievement emotions operate in university students is highly relevant, since these emotions have a positive or negative effect on students’ performance and their use of available personal resources. If a lack of positive emotionality and presence of negative emotionality mean that many behavioral resources must be applied, it is reasonable to think that fewer resources are left for task execution. Furthermore, burnout and lack of engagement will follow in the medium term, predisposing dropout or poor achievement [10]. 

A first implication for educational psychology would be the implementation of preventive programs that assess these variables in university students and determine what aspects stem from personal factors as opposed to contextual factors. With this knowledge, formative intervention processes could be designed for students and teachers, for the purpose of improving teaching and learning processes at the university [74], and to address the personal factor of perfectionism. 

Second, an understanding of the relationships presented here is essential to assessment and intervention in clinical and health psychology. These predictive models can be the foundation for specific strategies for treating the stress that characterizes university achievement. Achievement emotions, as well as other factors and symptoms of stress and burnout, should be accurately assessed, in the line of recent proposals [75]. From a Clinical Psychology approach, not only should individual variables be considered, but also their interaction with possibly relevant contextual variables, particularly in the academic setting. Among the contextual factors, common aspects should be identified, as well as aspects that explain the differential impact of achievement emotions in the three situations of classroom, study time and testing, as shown in this model [15]. Some of the factors that should be clinically evaluated are emotional self-regulation, certain personality variables (emotional stability, resilience, security, control), psychopathology, coping styles, emotional expression modes and social skills. On the other hand, interventions focused on modifying emotional regulation strategies could be designed to address aspects like activation or attention, or offer specific training to enhance performance. Such interventions could have an individual, group or online format, the latter showing demonstrated effectiveness in recent years [46,74].

## 10. Conclusions

Findings from this empirical study reveal that achievement emotions in fact have a linear, predictive relationship with the type of coping strategies applied, and ultimately, with the resulting motivational state. This linear relationship, however, seems to be influenced by the specific situation and by interaction with this situation, as shown in recent research [76,77]. Consequently, we must continue to investigate how the person x situation interaction is produced, in order to reach a better understanding of emotions during academic learning at university. The primary, secondary and tertiary prevention strategies that we use to provide psychological support to university students depend on this understanding, especially in cases where students are characterized by a maladjusted pattern of perfectionism. 

## Figures and Tables

**Figure 1 ijerph-17-02106-f001:**
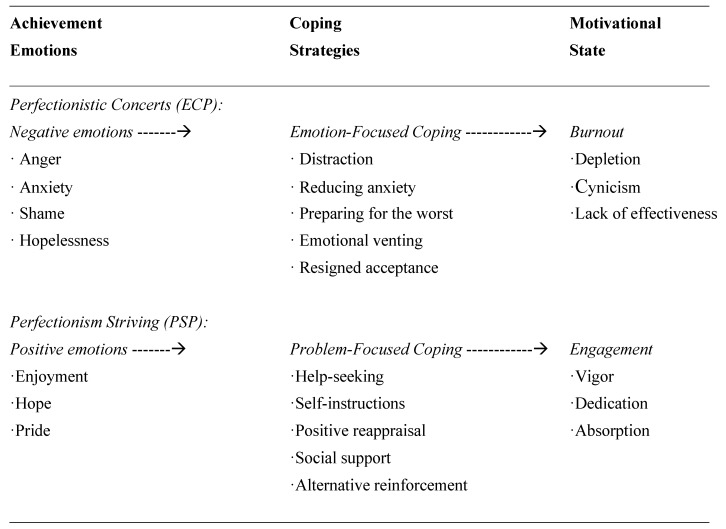
Relationship model between achievement emotions, coping strategies and engagement-burnout, based on the preview model [52] (pp. 151−152).

**Figure 2 ijerph-17-02106-f002:**
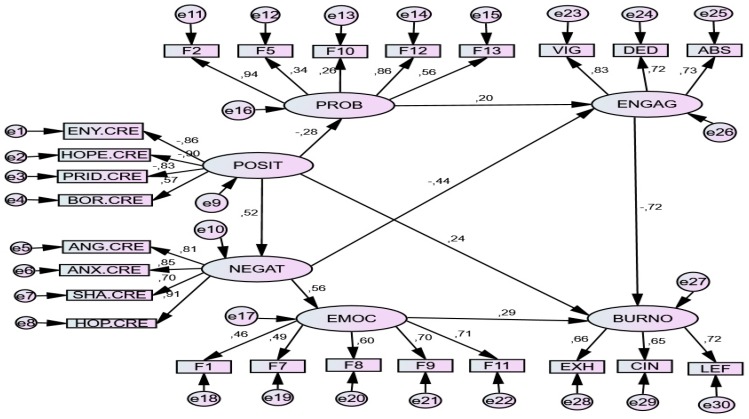
SEM model for Class Situation *(Stress level 1)*. *Note*. POSIT = *Positive Achievement Emotions* (Enjoyment, Hope, Pride); NEGAT = *Negative Achievement Emotions* (Anger, Anxiety, Shame, Hopelessness, Boredom); PROB. *Problem-focused strategies*: F2. Seeking help and family advice; F5. Self-Instructions; F10. Positive reappraisal and firmness; F12. Comunicating feelings and social support; F13. Seeking alternative reinforcement; EMOC = Emotion-focused strategies F1. Avoidant distraction; F7. Reducing anxiety and avoidance; F8. Preparing for the worst; F9. Emotional venting and isolation; F11. Resigned acceptance; ENGAG = Engagement (Vigor, Dedication, Absorption); BURNO = Burnout (Exhaustion, Cynicism, Lack of Effectiveness).

**Figure 3 ijerph-17-02106-f003:**
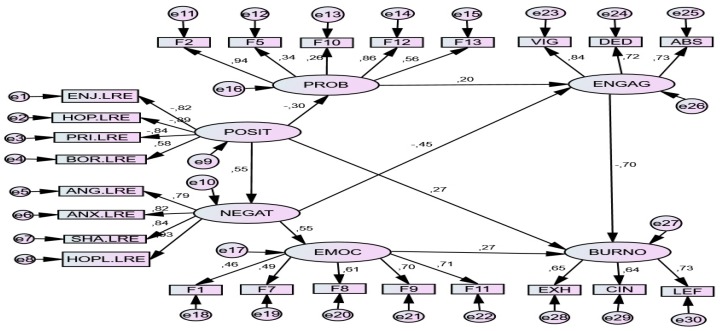
SEM model for Study Situation *(Stress level 2)*. *Note*. POSIT = *Positive Achievement Emotions* (Enjoyment, Hope, Pride); NEGAT = *Negative Achievement Emotions* (Anger, Anxiety, Shame, Hopelessness, Boredom); PROB. *Problem-focused strategies*: F2. Seeking help and family advice; F5. Self-Instructions; F10. Positive reappraisal and firmness; F12. Comunicating feelings and social support; F13. Seeking alternative reinforcement; EMOC = *Emotion-focused strategies* F1. Avoidant distraction; F7. Reducing anxiety and avoidance; F8. Preparing for the worst; F9. Emotional venting and isolation; F11. Resigned acceptance; ENGAG = *Engagement* (Vigor, Dedication, Absorption); BURNO = *Burnout* (Exhaustion, Cynicism, Lack of Effectiveness).

**Figure 4 ijerph-17-02106-f004:**
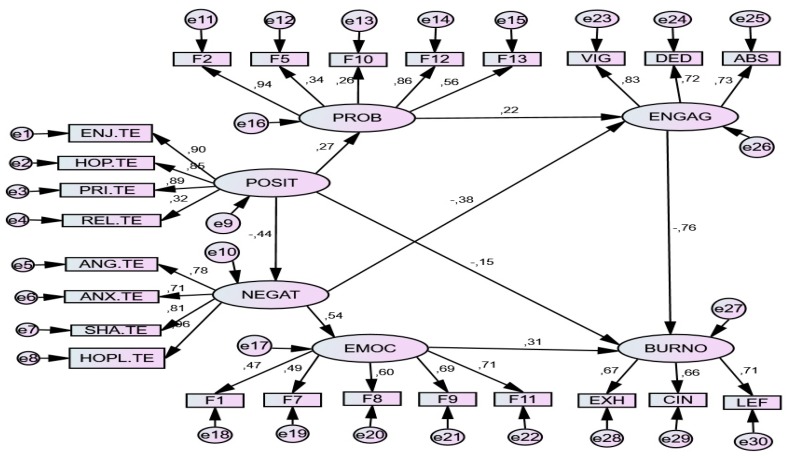
SEM model for Test Situation *(Stress level 3)*. *Note*. POSIT = *Positive Achievement Emotions* (Enjoyment, Hope, Pride); NEGAT = *Negative Achievement Emotions* (Anger, Anxiety, Shame, Hopelessness, Boredom); PROB. *Problem-focused strategies*: F2. Seeking help and family advice; F5. Self-Instructions; F10. Positive reappraisal and firmness; F12. Comunicating feelings and social support; F13. Seeking alternative reinforcement; EMOC = *Emotion-focused strategies* F1. Avoidant distraction; F7. Reducing anxiety and avoidance; F8. Preparing for the worst; F9. Emotional venting and isolation; F11. Resigned acceptance; ENGAG = *Engagement* (Vigor, Dedication, Absorption); BURNO = *Burnout* (Exhaustion, Cynicism, Lack of Effectiveness).

**Table 1 ijerph-17-02106-t001:** Types of Coping Strategies and Examples of Items in the Short EEC version [56].

***Emotion-focused coping* (D1)**	***Example items***
F1. Avoidant distraction	I sleep more than usual
F7. Reducing anxiety and avoidance	I decrease my anxiety by avoiding or escaping from situations that provoke it
F8. Preparing for the worst	I prepare myself for the worst
F9. Emotional venting and isolation	I act irritable and aggressive toward others
F11. Resigned acceptance	I accept the problem as it is, since I cannot do anything to solve it
***Problem-focused coping* (D2)**	***Example items***
F2. Seeking help and family advice	I ask a friend to help me clarify how I ought to tackle my problems
F5. Self-Instructions	I set down a plan of action and try to carry it out
F10. Positive reappraisal and firmness	I try to see positive aspects of the situation
F12. Comunicating feelings and social support	I feel better if I explain my problem to friends or family members
F13. Seeking alternative reinforcement	I start new activities (studies, etc.)

**Table 2 ijerph-17-02106-t002:** Linear regression coefficients (Beta) between *Achievement Emotions* (IV) and *Coping Strategies* (DV) with levels of stress (Class = level 1; Study = level 2; Test = level 3).

**Class**	**Positive**	**Negative**	**Enjoy**	**Hope**	**Pride**	**Boredom**	**Anger**	**Anxiety**	**Shame**	**Hopelessness**
Total	0.378 **	0.291 **		0.209 **	0.251 **					
D1	0.203 **	0.365 **				0.253 **	0.215 **			
D2	0.412 **			0.469 ***				0.220 **		
F1	0.136 *	0.246 **				0.168 *		0.205 **		
F7		0.171 *				0.206 **		0.202 **		−0.218 **
F8		0.341 **				0.185 **		0.259 **		
F9		0.417 ***		−0.242 **			0.200 **		0.158 **	
F11		0.353 **		−0.232 **					0.236 **	
F2	0.278 **			0.367 ***			−0.0192 *	0.233 **		
F5	0.358 ***			0.286 ***						−0.162 *
F10	0.423 **			0.510 ***						
F12	0.253 ***		−0.323 ***	0.321 **	0.246 **		−0.231 **	0.222 **		
F13				0.265 **	0.172 **					
**Study**	**Positive**	**Negative**	**Enjoy**	**Hope**	**Pride**	**Boredom**	**Anger**	**Anxiety**	**Shame**	**Hopelessness**
Total	0.502 ***	0.380 ***		0.203 *	0.193 *					
D1	0.283 ***	0.413 ***				0.221 **			0.211 **	
D2	0.515 ***	0.145 **		0.308 ***					0.183 *	−0.193 *
F1	0.124 *	0.246 ***								
F7		0.234 ***				0.252 **		0.235 **		
F8	0.141 **	0.417 ***			0.169 *				0.183 *	0.257 **
F9		0.419 ***								0.330 ***
F11		0.333 **		−0.159 *						0.475 ***
F2	0.370 **	0.213 *			0.163 *	0.165 *	−0.228 **	0.205 *	0.175 *	
F5	0.491 ***			0.120 *	0.235 **					
F10	0.434 ***			0.192 **	0.404 ***					
F12	0.332 **					0.249 **	−0.185 *		0.204 **	
F13	0.378 ***	0.245 **		0.344 **					0.185 *	
**Testing**	**Positive**	**Negative**	**Enjoy**	**Hope**	**Pride**	**Relief**	**Anger**	**Anxiety**	**Shame**	**Hopelessness**
Total	0.348 ***	0.255 ***		0.268 **	0.205 *		0.214 *			
D1	0.161 *	0.289 ***								
D2	0.385 ***				0336 ***	0.265 **	0.231 **			
F1		0.183 ***					0.172 *			
F7		0.147 **					−0.142 *			
F8		0.308 ***					0.186 *			
F9		0.331 ***	0.229 **				0.176 *			
F11	−0.118*	0.334 ***			−0.260 **			0.175 *		
F2	0.212 **	0.113 *			0.222 *			0.196 **		
F5	0.334 ***			0.380 ***						
F10	0.336 ***	0.110 *		0.419 ***				−0.252 **		
F12	0.219 **	0.115 *	−0.169 *	0.179 *	0.285 **			0.269 **		
F13	0.284 ***	0.187 *		0.201 *				0.146 *		

Note: *Emotion-focused coping (D1)*: F1. Avoidant distraction; F7. Reducing anxiety and avoidance; F8. Preparing for the worst; F9. Emotional venting and isolation; F11. Resigned acceptance; *Problem-focused coping* (D2): F2. Seeking help and family advice; F5. Self-Instruction; F10. Positive reappraisal and firmness; F12. Communicating feelings and social support; F13. Seeking alternative reinforcement. * *p* < 0.05; ** *p* < 0.01; *** *p* < 0.001.

**Table 3 ijerph-17-02106-t003:** Models of structural linear results of the variables.

Chi^2^	FG	*p <*	NFI	RFI	IFI	TLI	CFI	HOELT.	RMSEA
Model 4229.258 (324-382): 242		0.000	0.799	0.826	0.811	0.840	0.810	0.189	0.103
Model 4417.851 (324-380): 224		0.000	0.908	0.913	0.907	0.926	0.906	0.206	0.085

**Table 4 ijerph-17-02106-t004:** Standardized Direct Effects (Default model). Class situation.

	POSIT	NEGAT	PROB	EMOT	ENGAGEMENT	BURNOUT
NEGATIVE	0.525					
PROBLEM	−0.279					
EMOTION		0.562				
ENGAGEMENT		−0.444	0.200			
BURNOUT	0.237			0.290	-0.720	
ENJOYMENT	−0.856					
HOPE	−0.900					
PRIDE	−0.827					
BOREDOM	0.569					
ANGER		0.809				
ANXIETY		0.849				
SHAME		0.701				
HOPELESSNESS		0.910				
EEF2			0.939			
EEF5			0.337			
EECF10			0.258			
EECF12			0.862			
EECF13			0.563			
EECF1				0.462		
EEFC7				0.485		
EECF8				0.599		
EECF9				0.700		
EECF11				0.713		
VIGOR					0.829	
DEDICATION					0.722	
ABSORPTION					0.730	
EXHAUSTION						0.664
CYNICISM						0.655
LACK OF EFFECTIVENESS						0.717

*Note:* (D2) *Problem-focused coping*: F2. Seeking help and family advice; F5. Self-Instruction; F10. Positive reappraisal and firmness; F12. Communicating feelings and social support; F13. Seeking alternative reinforcement; (D1) *Emotion-focused coping*: F1. Avoidant distraction; F7. Reducing anxiety and avoidance; F8. Preparing for the worst; F9. Emotional venting and isolation; F11. Resigned acceptance.

**Table 5 ijerph-17-02106-t005:** Standardized Indirect Effects (Default model). Class Situation.

	POSIT	NEGAT	PROB	EMOT	ENGAGEMENT	BURNOUT
NEGATIVE						
PROBLEM						
EMOTION	0.295					
ENGAGEMENT	−0.288					
BURNOUT	0.292	0.482	−0.143			
ENJOYMENT						
HOPE						
PRIDE						
BOREDOM						
ANGER	0.424					
ANXIETY	0.445					
SHAME	0.367					
HOPELESSNESS	0.477					
EEF2	−0.262					
EEF5	−0.094					
EECF10	−0.072					
EECF12	−0.241					
EECF13	−0.157					
EECF1	0.136					
EEFC7	0.143					
EECF8	0.177					
EECF9	0.206					
EECF11	0.210					
VIGOR	−0.239	−0.368	0.165			
DEDICATION	−0.208	−0.320	0.144			
ABSORPTION	−0.211	−0.324	0.146			
EXHAUSTION	0.352	0.320	−0.095	0.193	−0.476	
CYNICISM	0.347	0.315	−0.093	0.191	−0.469	
LACK OF EFFECTIVENESS	0.380	0.345	−0.102	0.209	−0.514	

*Note:* (D2) *Problem-focused coping*: F2. Seeking help and family advice; F5. Self-Instruction; F10. Positive reappraisal and firmness; F12. Communicating feelings and social support; F13. Seeking alternative reinforcement; (D1) *Emotion-focused coping*: F1. Avoidant distraction; F7. Reducing anxiety and avoidance; F8. Preparing for the worst; F9. Emotional venting and isolation; F11. Resigned acceptance.

**Table 6 ijerph-17-02106-t006:** Models of structural linear results of the variables.

Chi^2^	FG	*p <*	NFI	RFI	IFI	TLI	CFI	HOELT.	RMSEA
Model 4016.804 (299-81): 218		0.000	0.809	0.758	0.876	0.839	0.873	0.169	0.078
Model 4257.872 (324-380): 224		0.000	0.906	0.927	0.907	0.940	0.908	0.204	0.080

**Table 7 ijerph-17-02106-t007:** Standardized Direct Effects (Default model): Study Situation.

	POSIT	NEGAT	PROB	EMOT	ENGAGEMENT	BURNOUT
NEGATIVE	0.555					
PROBLEM	−0.301					
EMOTION		0.552				
ENGAGEMENT		−0.454	0.204			
BURNOUT	0.268			0.269	−0.705	
ENJOYMENT	−0.822					
HOPE	−0.889					
PRIDE	−0.836					
BOREDOM	0.580					
ANGER		0.793				
ANXIETY		0.820				
SHAME		0.843				
HOPELESSNESS		0.930				
EEF2			0.937			
EEF5			0.338			
EECF10			0.259			
EECF12			0.863			
EECF13			0.563			
EECF1				0.463		
EEFC7				0.489		
EECF8				0.605		
EECF9				0.696		
EECF11				0.712		
VIGOR					0.838	
DEDICATION					0.716	
ABSORPTION					0.734	
EXHAUSTION						0.654
CYNICISM						0.644
LACK OF EFFECTIVENESS						0.728

*Note:* (D2) *Problem-focused coping*: F2. Seeking help and family advice; F5. Self-Instruction; F10. Positive reappraisal and firmness; F12. Communicating feelings and social support; F13. Seeking alternative reinforcement; (D1) *Emotion-focused coping*: F1. Avoidant distraction; F7. Reducing anxiety and avoidance; F8. Preparing for the worst; F9. Emotional venting and isolation; F11. Resigned acceptance.

**Table 8 ijerph-17-02106-t008:** Standardized Indirect Effects (Default model). Learning Situation.

	POSIT	NEGAT	PROB	EMOT	ENGAGEMENT	BURNOUT
NEGATIVE						
PROBLEM						
EMOTION	0.306					
ENGAGEMENT	−0.134					
BURNOUT	0.306	0.469	−0.144			
ENJOYMENT						
HOPE						
PRIDE						
BOREDOM						
ANGER	0.440					
ANXIETY	0.455					
SHAME	0.468					
HOPELESSNESS	0.516					
EEF2	−0.282					
EEF5	−0.102					
EECF10	−0.078					
EECF12	−0.260					
EECF13	−0.170					
EECF1	0.142					
EEFC7	0.150					
EECF8	0.185					
EECF9	0.213					
EECF11	0.218					
VIGOR	−0.263	−0.381	0.171			
DEDICATION	−0.225	−0.325	0.146			
ABSORPTION	−0.230	−0.334	0.150			
EXHAUSTION	0.373	0.306	−0.094	0.176	−0.461	
CYNICISM	0.368	0.302	−0.093	0.173	−0.454	
LACK OF EFFECTIVENESS	0.416	0.341	−0.105	0.196	−0.513	

*Note:* (D2) *Problem-focused coping*: F2. Seeking help and family advice; F5. Self-Instruction; F10. Positive reappraisal and firmness; F12. Communicating feelings and social support; F13. Seeking alternative reinforcement; (D1) *Emotion-focused coping*: F1. Avoidant distraction; F7. Reducing anxiety and avoidance; F8. Preparing for the worst; F9. Emotional venting and isolation; F11. Resigned acceptance.

**Table 9 ijerph-17-02106-t009:** Models of structural linear results of the variables.

Chi^2^	FG	*p <*	NFI	RFI	IFI	TLI	CFI	HOELT.	RMSEA
Model 2544.602 (299-81): 218		0.000	0.823	0.855	0.816	0.809	0.856	0.169	0.078
Model 3900.927 (324-380): 224		0.000	0.905	0.937	0.918	0.925	0.917	0.203	0.080

**Table 10 ijerph-17-02106-t010:** Standardized Direct Effects (Default model): Test Situation.

	POSIT	NEGAT	PROB	EMOT	ENGAGEMENT	BURNOUT
NEGATIVE	−0.439					
PROBLEM	0.273					
EMOTION		0.538				
ENGAGEMENT		−0.378	0.225			
BURNOUT	−0.151			0.313	−0.757	
ENJOYMENT	0.903					
HOPE	0.850					
PRIDE	0.891					
BOREDOM	0.322					
ANGER		0.784				
ANXIETY		0.707				
SHAME		0.807				
HOPELESSNESS		0.956				
EEF2			0.938			
EEF5			0.337			
EECF10			0.258			
EECF12			0.863			
EECF13			0.564			
EECF1				0.466		
EEFC7				0.491		
EECF8				0.604		
EECF9				0.691		
EECF11				0.712		
VIGOR					0.831	
DEDICATION					0.721	
ABSORPTION					0.732	
EXHAUSTION						0.668
CYNICISM						0.657
LACK OF EFFECTIVENESS						0.714

*Note:* (D2) *Problem-focused coping*: F2. Seeking help and family advice; F5. Self-Instruction; F10. Positive reappraisal and firmness; F12. Communicating feelings and social support; F13. Seeking alternative reinforcement; (D1) *Emotion-focused coping*: F1. Avoidant distraction; F7. Reducing anxiety and avoidance; F8. Preparing for the worst; F9. Emotional venting and isolation; F11. Resigned acceptance.

**Table 11 ijerph-17-02106-t011:** Standardized Indirect Effects (Default model). Test Situation.

	POSIT	NEGAT	PROB	EMOT	ENGAGEMENT	BURNOUT
NEGATIVE						
PROBLEM						
EMOTION	−0.236					
ENGAGEMENT	0.227					
BURNOUT	−0.246	0.454	−0.170			
ENJOYMENT						
HOPE						
PRIDE						
BOREDOM						
ANGER	−0.334					
ANXIETY	−0.310					
SHAME	−0.354					
HOPELESSNESS	−0.420					
EEF2	0.257					
EEF5	0.092					
EECF10	0.070					
EECF12	0.236					
EECF13	0.154					
EECF1	−0.110	0.251				
EEFC7	−0.116	0.264				
EECF8	−0.143	0.325				
EECF9	−0.163	0.372				
EECF11	−0.168	0.383				
VIGOR	0.166	−0.314	0.187			
DEDICATION	0.164	−0.273	0.162			
ABSORPTION	0.189	−0.276	0.164			
EXHAUSTION	−0.265	0.304	−0.170	0.209	−0.506	
CYNICISM	−0.281	0.298	−0.114	0.205	−0.497	
LACK OF EFFECTIVENESS	−0.263	0.324	−0.112	0.233	−0.540	

*Note:* (D2) *Problem-focused coping*: F2. Seeking help and family advice; F5. Self-Instruction; F10. Positive reappraisal and firmness; F12. Communicating feelings and social support; F13. Seeking alternative reinforcement; (D1) *Emotion-focused coping*: F1. Avoidant distraction; F7. Reducing anxiety and avoidance; F8. Preparing for the worst; F9. Emotional venting and isolation; F11. resignedacceptance.

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
