# Peer review of "A Structural Equation Model of Achievement Emotions, Coping Strategies and Engagement-Burnout in Undergraduate Students: A Possible Underlying Mechanism in Facets of Perfectionism"

_ijerph, 2020, doi:10.3390/ijerph17062106_

Round 1

Reviewer 1 Report

There are some good ideas in this study, however the presentation and writing of the manuscript require significant additional thought. It is rather difficult to follow and understand at the moment.

Specific comments:

Please change "The test hypothesis stated that" to "The test hypothesis was that". Please change "All students granted informed consent" to "All students provided informed consent". Please define the abbreviation 'SEM' in the first instance of its use. Please change "engagement vs burnout" to "engagement versus burnout". The concept of 'valence' can be further explained in the introduction. It should be stated that engagement is the hypothesized opposite of burnout. How was sample size determined? There is currently no evidence of power calculation. The present sample appears limited to a convenience sample. Why did the authors not administer the Maslach Burnout Inventory (MBI)? This is validated by extensive research and widely applied. Please provide the actual IRB study/approval number. It is unclear if study participation was incentivized in any way, and was anonymity of the students ensured? On average, how long did it take a student to complete the online questionnaire? As the questionnaires appear rather extensive, how did the study investigators deal with the issue of respondent fatigue? The statistical analyses could be better presented. There are two diagrams labelled as Figure 1.

Reviewer 2 Report

Review for the manuscript:

“Achievement Emotions, Coping Strategies and Engagement-Burnout in University Undergraduates: a SEM model”

The manuscript examines linear and predictive relations between university students’ achievement emotions, coping strategies, and engagement-burnout in three learning situations: the classroom, study time, and testing. The authors present the results of logistic regression analyses and structural equation models to test their hypotheses. Strengths of the manuscript include comprehensive analyses and research into an important topic that warrants further attention. Nevertheless, certain aspects should be considered to improve the quality of the manuscript.

Conceptual Suggestions:

Introduction

The authors state that “research seeks to explain to what degree emotional processes facilitate or interfere in the learning process”, and then proceed to describe academic stress. While I can appreciate that academic stress is strongly intertwined with the research topic at hand and warrants attention in this paper, I would have expected that the authors focus more strongly in the initial introduction section on the importance of achievement emotions, and how coping mechanisms and engagement/burnout are theoretically intertwined. As a result of this point, the final sentence of this paragraph introduces the variables of interest for the current study rather abruptly and in my opinion, could be more clearly prepared in the preceding sentences. I would suggest that the authors integrate the role/relevance of their variables of interest in the current study (achievement emotions, coping strategies, engagement/burnout) in a more clear and obvious manner. Moreover, a more specific and justifying rationale should be provided in the introduction for looking into the predictive relationship between “achievement emotions and coping strategies, as well as the effect of these two variables on the motivational state of engagement vs burnout in university students”, above and beyond the fact that this topic has not yet been researched yet. It would be helpful for the reader if the authors already mentioned and briefly explain the three situations of interest for this study and why they are specifically important (classroom, study time, testing) – as a general note, the authors should also mention these situations and how they might relate to the constructs described in the following theoretical sections (as it currently stands, the authors first make mention of these three situations in their aims/hypotheses section, which is rather abrupt and not sufficiently theoretically prepared).

Achievement Related Emotions as an affective variable of the learning process  

The authors describe literature surrounding achievement emotions in the university context. Here, I would suggest that the authors describe more clearly and specifically the effects found for particular emotions and take care to focus on literature which examines the emotions which are being measured in the current study. For example, when the authors state “Achievement emotions are reported to be interrelated with meta cognitive monitoring processes”, it would strengthen their argumentation and be helpful for the reader to know which particular achievement emotions the author is referring to. The same holds true for: “positive emotions to have a positive predictive value” (which positive emotions, a predictive value of what variable?). This should be considered for all descriptions of study results concerning emotions throughout the paper. Please make mention of how achievement related emotions are related to the three situations that you initially described (classroom, study time, testing), also in the next sections which describe the other variables of interest.

Engagement-Burnout as a motivational variable of the learning process

Although the authors state at the end of this section that the relationship of burnout and engagement with university students’ achievement emotions and coping strategies has yet to be explicitly established, literature exists to support certain aspects of this relationship (for example, emotions as predictors of burnout or engagement) which can be mentioned here to better link the constructs and expected relations. Moreover, rationale needs to be provided concerning why the burnout-engagement link is important to research (again, past the point that it hasn’t yet been researched).

Aims and Hypotheses

Figure 1

This model is comprehensive and helpful but it should be better prepared from a theoretical standpoint in the previous sections. While the authors previously explained and provided sufficient rationale in their theoretical background regarding examining the particular emotions of enjoyment, hope, pride, anger, anxiety, shame, hopelessness, boredom, and relief, they did not do the same for the particular coping strategies that they examined. The authors should explain the specific problem focused and emotion focused coping strategies that they will examine already in their theoretical background section, as well as some rationale as to why they expect these should be tied to the specific emotions they present in this model. In general, it would also be helpful if the authors explained why they expect that all positive and all negative emotions should function in the same way when it comes to problem and emotion focused coping strategies (i.e., why is it not the case that a particular discrete emotion, such as enjoyment, might facilitate the use of a different coping strategy than the emotion of hope? Why is it that the distinct positive emotions should be conflated to the level of general ‘positive emotions’ when looking into relations with coping strategies?).

Methods and Analyses

The authors should be careful not to lead the readers to assume that this study allowed for insight into causality and temporal trends, as the study was correlational in nature. In the participants section, please explain if there was missing data, and if so, how it was handled. In the instruments section, it is somewhat confusing that the AEQ is initially described, followed by the other scales, and then the AEQ is again discussed a final time. In the procedure section, the word ‘platform’ is italicized, why is this? Was there a particular platform that was used by de la Fuente et al., 2015 in their study which is of importance for the current study? If so, please describe what this is and why it is important.

Results

In the linear predictive relationships section, the distinction between the three situations (classroom, study, and testing) finally become clearer, and it is helpful that this distinction is now made in an organized way, but as stated previously, the importance and rationale of these particular situations should be theoretically prepared. It would also be helpful if the authors explained more thoroughly what exactly constitutes as being burnt out or engaged in the particular scales that the authors used. In Table 2, please double check the formatting, for example, a decimal is incorrectly underlined under the TLI column. Please make sure that the tables are in APA format and that the standard errors are described regarding the SEM results. It would be helpful for the authors to include a table containing the bivariate correlations between all variables of interest.

Limitations and future research

The correlational design of the current study should be mentioned as a limitation. As a general comment, I was able to find errors regarding grammar and wording, please read through the manuscript again to correct any errors.

Author Response

The manuscript examines linear and predictive relations between university students’ achievement emotions, coping strategies, and engagement-burnout in three learning situations: the classroom, study time, and testing. The authors present the results of logistic regression analyses and structural equation models to test their hypotheses. Strengths of the manuscript include comprehensive analyses and research into an important topic that warrants further attention. Nevertheless, certain aspects should be considered to improve the quality of the manuscript.

Conceptual Suggestions:

Introduction

The authors state that “research seeks to explain to what degree emotional processes facilitate or interfere in the learning process”, and then proceed to describe academic stress. While I can appreciate that academic stress is strongly intertwined with the research topic at hand and warrants attention in this paper, I would have expected that the authors focus more strongly in the initial introduction section on the importance of achievement emotions, and how coping mechanisms and engagement/burnout are theoretically intertwined. As a result of this point, the final sentence of this paragraph introduces the variables of interest for the current study rather abruptly and in my opinion, could be more clearly prepared in the preceding sentences. I would suggest that the authors integrate the role/relevance of their variables of interest in the current study (achievement emotions, coping strategies, engagement/burnout) in a more clear and obvious manner. Moreover, a more specific and justifying rationale should be provided in the introduction for looking into the predictive relationship between “achievement emotions and coping strategies, as well as the effect of these two variables on the motivational state of engagement vs burnout in university students”, above and beyond the fact that this topic has not yet been researched yet. It would be helpful for the reader if the authors already mentioned and briefly explain the three situations of interest for this study and why they are specifically important (classroom, study time, testing) – as a general note, the authors should also mention these situations and how they might relate to the constructs described in the following theoretical sections (as it currently stands, the authors first make mention of these three situations in their aims/hypotheses section, which is rather abrupt and not sufficiently theoretically prepared).

 RESPONSE: Thank you. The comment is very pertinent. The structure of the introduction has been revised, establishing more clearly:

(1) the relationships between emotions, coping strategies and engagement-burnout

(2) the relevance of each situation, as an example of a stress context, to analyze the factor invariance, as well as the effect of the differences in each situation in the relationship studied.

(3) the potential realization with the perfectionism construct, not previously finished. Although it is not the object of this research to integrate this relationship to establish differences between types of perfectionism.

Achievement Related Emotions as an affective variable of the learning process  

The authors describe literature surrounding achievement emotions in the university context. Here, I would suggest that the authors describe more clearly and specifically the effects found for particular emotions and take care to focus on literature which examines the emotions which are being measured in the current study. For example, when the authors state “Achievement emotions are reported to be interrelated with meta cognitive monitoring processes”, it would strengthen their argumentation and be helpful for the reader to know which particular achievement emotions the author is referring to. The same holds true for: “positive emotions to have a positive predictive value” (which positive emotions, a predictive value of what variable?). This should be considered for all descriptions of study results concerning emotions throughout the paper. Please make mention of how achievement related emotions are related to the three situations that you initially described (classroom, study time, testing), also in the next sections which describe the other variables of interest.

 RESPONSE: Thank you. Section information has been reviewed.

Engagement-Burnout as a motivational variable of the learning process

Although the authors state at the end of this section that the relationship of burnout and engagement with university students’ achievement emotions and coping strategies has yet to be explicitly established, literature exists to support certain aspects of this relationship (for example, emotions as predictors of burnout or engagement) which can be mentioned here to better link the constructs and expected relations. Moreover, rationale needs to be provided concerning why the burnout-engagement link is important to research (again, past the point that it hasn’t yet been researched).

RESPONSE. Thank you. It is true. New evidence has been reviewed and inserted

Aims and Hypotheses

Figure 1

This model is comprehensive and helpful but it should be better prepared from a theoretical standpoint in the previous sections. While the authors previously explained and provided sufficient rationale in their theoretical background regarding examining the particular emotions of enjoyment, hope, pride, anger, anxiety, shame, hopelessness, boredom, and relief, they did not do the same for the particular coping strategies that they examined. The authors should explain the specific problem focused and emotion focused coping strategies that they will examine already in their theoretical background section, as well as some rationale as to why they expect these should be tied to the specific emotions they present in this model. In general, it would also be helpful if the authors explained why they expect that all positive and all negative emotions should function in the same way when it comes to problem and emotion focused coping strategies (i.e., why is it not the case that a particular discrete emotion, such as enjoyment, might facilitate the use of a different coping strategy than the emotion of hope? Why is it that the distinct positive emotions should be conflated to the level of general ‘positive emotions’ when looking into relations with coping strategies?).

 RESPONSE: Thank you. The information in the previous sections has been completed so that the model does not appear abruptly. The hypothetical relationship with the two types of perfectionism has been completed.

Methods and Analyses

The authors should be careful not to lead the readers to assume that this study allowed for insight into causality and temporal trends, as the study was correlational in nature. In the participants section, please explain if there was missing data, and if so, how it was handled. In the instruments section, it is somewhat confusing that the AEQ is initially described, followed by the other scales, and then the AEQ is again discussed a final time. In the procedure section, the word ‘platform’ is italicized, why is this? Was there a particular platform that was used by de la Fuente et al., 2015 in their study which is of importance for the current study? If so, please describe what this is and why it is important.

 RESPONSE: Thanks. Text adjustments have been made.

Results

In the linear predictive relationships section, the distinction between the three situations (classroom, study, and testing) finally become clearer, and it is helpful that this distinction is now made in an organized way, but as stated previously, the importance and rationale of these particular situations should be theoretically prepared. It would also be helpful if the authors explained more thoroughly what exactly constitutes as being burnt out or engaged in the particular scales that the authors used.

In Table 2, please double check the formatting, for example, a decimal is incorrectly underlined under the TLI column.

Please make sure that the tables are in APA format and that the standard errors are described regarding the SEM results.

It would be helpful for the authors to include a table containing the bivariate correlations between all variables of interest.

 RESPONSE: Thank you, the format of the tables has been revised. It has also been explained previously that situations represent three different levels of academic stress: 1 = CLASS; 2 = STUDY; 3 = EXAM.

It is not considered relevant to insert a Correlation Table because these values are presented in the multiple regression Table.

Limitations and future research

The correlational design of the current study should be mentioned as a limitation. As a general comment, I was able to find errors regarding grammar and wording, please read through the manuscript again to correct any errors.

RESPONSE.  Thank you. Indeed, it is a limitation, which we have already inserted. The entire manuscript has been reviewed by a foreign translator from the USA.

Reviewer 3 Report

This paper requires extensive editing. The writing was convoluted and it was challenging to pick out the main findings of the study.

The sampling strategy requires more explanation. It appears that most of the respondents are females (n = 536, 83.5%). Authors should in their analysis as independent variables sex and age in order to examine possible main effects and interaction effects.

It is unclear what this paper is adding to the existing literature.

Author Response

This paper requires extensive editing. The writing was convoluted and it was challenging to pick out the main findings of the study.

RESPONSE. The report has been reviewed in depth, in depth and form.

The sampling strategy requires more explanation. It appears that most of the respondents are females (n = 536, 83.5%). Authors should in their analysis as independent variables sex and age in order to examine possible main effects and interaction effects.

RESPONSE: The analysis of GENDER and AGE deviates from the objective of the article analysis. It has been inserted as a limitation and necessary analysis for future research, as modulatory variables of the effects found. Thank you.

It is unclear what this paper is adding to the existing literature.

RESPONSE. This information has already been inserted in the report.

Round 2

Reviewer 1 Report

  1. The length of the paper and number of references is rather extensive and could be trimmed. It is important to be succinct in your writing.

Author Response

Thanks for the improvement suggestions.

1) The article has been revised and reduced to 11,894 words. Initially it had 13335 words

2) Citations and references have been selected. It has been reduced to a total of 97. Initially there were 134 references.
